

# Differential scanning calorimetry of whole *Escherichia coli* treated with the antimicrobial peptide MSI-78 indicate a multi-hit mechanism with ribosomes as a novel target

Alexander M. Brannan[1], William A. Whelan[1], Emma Cole[1] and Valerie Booth[1,2]

[1] Department of Biochemistry, Memorial University of Newfoundland, St. John's Newfoundland and Labrador, Canada

[2] Department of Physics and Physical Oceanography, Memorial University of Newfoundland, St. John's Newfoundland and Labrador, Canada

## ABSTRACT

Differential Scanning Calorimetry (DSC) of intact *Escherichia coli (E. coli)* was used to identify non-lipidic targets of the antimicrobial peptide (AMP) MSI-78. The DSC thermograms revealed that, in addition to its known lytic properties, MSI-78 also has a striking effect on ribosomes. MSI-78's effect on DSC scans of bacteria was similar to that of kanamycin, an antibiotic drug known to target the 30S small ribosomal subunit. An *in vitro* transcription/translation assay helped confirm MSI-78's targeting of ribosomes. The scrambled version of MSI-78 also affected the ribosome peak of the DSC scans, but required greater amounts of peptide to cause a similar effect to the unscrambled peptide. Furthermore, the effect of the scrambled peptide was not specific to the ribosomes; other regions of the DSC thermogram were also affected. These results suggest that MSI-78's effects on *E. coli* are at least somewhat dependent on its particular structural features, rather than a sole function of its overall charge and hydrophobicity. When considered along with earlier work detailing MSI-78's membrane lytic properties, it appears that MSI-78 operates via a multi-hit mechanism with multiple targets.

## INTRODUCTION

AMPs are a crucial component of the innate immune system of many organisms and can protect against a variety of invading pathogens including bacteria, viruses, and microbial eukaryotes (*Shai, 1999*; *Zasloff, 2002*; *Jenssen, Hamill & Hancock, 2006*; *Nguyen, Haney & Vogel, 2011*; *Wimley & Hristova, 2011*; *Bechinger & Salnikov, 2012*). AMPs have been isolated from a plethora of organisms, including plants, humans, frogs, and bacterial species. They are generally short, typically ranging from 12 to 50 amino acids in length, cationic, due to an abundance of basic residues (lysine, arginine, histidine), and amphipathic, with some AMPs displaying a hydrophobicity of >50%. These characteristics are important in

Corresponding author
Valerie Booth, vbooth@mun.ca

driving the interactions of AMPs with the membrane of target pathogens, which can result in the direct killing of the pathogen via membrane disruption, and/or provide a means for the AMP to access intracellular targets.

Much hope has been expressed for the potential of novel AMP-based therapeutics to help address the growing problem of antibiotic resistance to conventional small antibiotics (*Rossi et al., 2008*; *Spellberg et al., 2008*; *Boucher et al., 2009*; *Yeung, Gellatly & Hancock, 2011*). However, successfully developing AMP-based drugs, in particular for internal use, has proved very challenging (*Matsuzaki, 2009*; *Wimley & Hristova, 2011*; *Pasupuleti, Schmidtchen & Malmsten, 2012*). One major barrier to the optimization of AMPs for therapeutic use rests with the simplified model membrane systems in which detailed mechanistic studies and AMP sequence optimization are generally performed. By focusing solely on the lipids, studies that are limited to model membrane systems fail to capture the myriad of potentially important interactions that AMPs may have with their target microbes.

An example of the gap that exists between mechanistic studies in model membranes and the conditions under which AMPs actually carry out their function is the observation that AMPs cause leakage in model membranes at protein:lipid (P:L) ratios $\sim$10,000 times lower than the P:L ratio needed to see bacterial growth inhibition (*Wimley, 2010*). On the one hand, solid state NMR, vesicle leakage and other biophysical experiments aimed at understanding the structural mechanisms of AMP-induced bilayer disruptions employ model membranes and overall AMP:lipid molar ratios of $\sim$1: 200 (*Wimley, 2010*). On the other hand, the minimal inhibitory concentration (MIC) assays used to define the concentration of AMP needed to prevent bacterial growth are typically performed with overall molar ratios of $\sim$1000 AMP:1 lipid—a difference in AMP:lipid ratio of $\sim$5 $\times$ 10$^6$ compared to the biophysical studies. Furthermore, the hemolytic assays that are frequently used to indicate host cell toxicity are done at orders of magnitude higher cell densities than MIC assays (*Matsuzaki, 2009*; *Wimley, 2010*). This results in very different P:L ratios (e.g., $\sim$1:15 for our hemolytic assay at 24.35 ug/mL MSI-78) of 3 to 4 orders of magnitude less AMP per lipid as compared to the MIC assays. Beyond these very disparate experimental protocols, it should also not be forgotten that *in vivo*, there will be a competition between AMP-pathogen binding and AMP-host cell binding that may alter the effective selectivity of an AMP for the pathogen (*Bagheri, Taheri-Araghi & Ha, 2015*). Attempts to make sense of these disparate experimental conditions have been made by working out a more "local" (to the membrane) AMP concentration. For example, Melo, Ferre and Castanho (*Melo, Ferre & Castanho, 2009*) took the approach of assuming that the AMP would bind only the lipid component of the bacteria and used known partition constants to suggest that AMP:lipid ratios of 1:16 at the membrane could be achieved for the pathogen densities expected for *in vivo* conditions. Alternatively, Wimley (*Wimley, 2010*), estimated how much AMP would actually bind to the bacteria in a typical MIC experiment and came up with 100:1 for the bound AMP:lipid ratio. From this he concluded that much of the AMP **must** bind to non-lipidic cell components to account for the discrepancy between the amounts of peptide needed to see disruption in model membranes versus bacteria. Even the relative leakage induced in model lipid membranes

by different AMPs does not always correlate with their relative ability to prevent bacterial growth (*Friedrich et al., 2000*; *He, Krauson & Wimley, 2014*).

Thus, while AMP-lipid interactions are key in understanding AMP mechanism, it is vital to look beyond model membranes and examine AMPs' potentially functional interactions with other components of the target cell. While the vast majority of studies of AMP mechanism have focused on lipids, there are indeed AMPs for which non-lipidic intracellular targets are known (*Brogden, 2005*; *Marcos & Gandia, 2009*), although AMP-membrane interactions can still be important for allowing AMPs to access the interior of the cell. Known non-lytic mechanisms for AMPs include targeting nucleic acids (*Park, Kim & Kim, 1998*),chaperones (*Kragol et al., 2001*), cell wall synthesis (*Brotz et al., 1998*), protein synthesis (*Patrzykat et al., 2002*), and enzymatic activity (*Andreu & Rivas, 1998*), as well as the G-quadruplex in mammalian cancer cells (*Jana et al., 2013*).

In order to provide a more integrated view of AMP-bacteria interactions, one strategy is to start with the same biophysical methods used to study and optimize AMP inter-actions with model membrane systems, and extend these methods for use with whole bacteria (e.g., *Gee et al., 2013*; *Maisetta et al., 2013*; *Hall et al., 2014*). In this work, we describe differential scanning calorimetry (DSC) studies of the AMP MSI-78 interacting with whole bacteria.

MSI-78 is a well-studied and clinically tested analogue of magainin-2 (*Zasloff, 1987*; *Gottler & Ramamoorthy, 2009*). It has 22 residues and possesses a +9 net charge, due to the abundance of lysine residues, as well as a large hydrophobic content (55%). Solid state NMR and other biophysical studies indicate that MSI-78 induces positive curvature strain and forms torroidal pores in model lipid bilayer systems (*Hallock, Lee & Ramamoorthy, 2003*; *Ramamoorthy et al., 2006*; *Ramamoorthy, 2009*; *Lee et al., 2013*; *Lee et al., 2015*; *Porcelli et al., 2006*). The presence of MSI-78 leads to a blue shift in the fluorescence emission maximum of ANS (8-Anilino-1-naphthalenesulfonic acid) treated *E. coli* cells and also causes the release of fluorescent dye from lipid vesicles (*Ramamoorthy et al., 2006*), which together suggest that interactions with both the inner and outer membranes are involved in MSI-78's mechanism. Previous work in our lab (*Pius, Morrow & Booth, 2012*) extended NMR studies of MSI-78 beyond model lipid systems to whole *E. coli*, using solid state NMR. This study suggested (1) a large fraction of MSI-78 does not reach the bilayer and may remain in association with non-lipidic components of the cell envelope and (2) even at relatively low levels of MSI-78, there is a large degree of membrane disruption, suggesting that the peptide may be able to reach the inside of the cells and interact with an intracellular component. Thus, to further probe the interaction of MSI-78 with intact bacteria, we turned to DSC, a method that, in principle, can be used to simultaneously reveal the effect of the peptide on lipid and non-lipidic cell envelope components, as well as intracellular components, of intact *E. coli*.

Whole cell differential scanning calorimetry (DSC) has been used to characterize the thermal stability of cell components in bacterial cells (*Miles, Mackey & Parsons, 1986*; *Mackey et al., 1988*; *Lepock, Frey & Inniss, 1990*; *Anderson et al., 1991*; *Anderson et al., 1991?*; *Mackey et al., 1991*; *Mackey et al., 1993*; *Teixeira et al., 1997*; *Mohácsi-Farkas et al., 1999*; *Bayles et al., 2000*; *Lee & Kaletunc, 2002a*; *Lee & Kaletunc, 2002b*; *Alpas et al.,*
*2003*; *Kaletunc et al., 2004*; *Lepock, 2005*; *Nguyen, Corry & Miles, 2006*; *Tunick, Bayles & Novak, 2006*; *Abuladze et al., 2009*; *Lee & Kaletunç, 2010*), archaeal cells (*Milek et al., 2007*), endospores (*Belliveau et al., 1992*), and eukaryotic cells (*Lepock et al., 1990*; *Obuchi et al., 2000*). DSC has also been used to probe the mechanisms of numerous AMPs in model membrane systems, including the cathelicidins (*Andrushchenko, Vogel & Prenner, 2007*), LL-37 (*Sevcsik et al., 2007*), β-17 (*Epand et al., 2003*), and MSI-78 (*Hallock, Lee & Ramamoorthy, 2003*; *Ramamoorthy et al., 2006*). The aim of our study was to use DSC to reveal the interactions of MSI-78 with components of whole cells, in particular with the non-lipidic cell envelope and intracellular cell components.

## MATERIALS AND METHODS

### Peptide and bacteria preparation

MSI-78 (*Maloy & Kari, 1995*) and a scrambled version of MSI-78 were chemically synthesized, purified by HPLC, and desalted, as in (*Pius, Morrow & Booth, 2012*). Stocks of 60 μL *E. coli* JM109 (1:1-culture/glycerol) were stored at −80 °C until use. 50 μL of culture was added to 10 mL of Luria-Bertani (LB) broth in 15 mL conical tubes and placed overnight in an incubator at 37.0 °C rotating at 160 rpm. 200 μL of this overnight culture was then used to inoculate 20 mL sterile LB broth in a 100 ml Erlenmeyer. This was grown, under the same conditions described above, until the bacterial culture reached mid-log phase as indicated by an optical density of approximately 0.65 when measured at a wavelength of 600 nm. Ten mL of this mid log culture was transferred to a 50 mL centrifuge tube and centrifuged at 4100 g, at 4 °C, for 10 min in a Thermo Sorvall RC6+ Centrifuge, using an f21s (6 × 50) rotor (Thermo Fisher Scientific Inc., Waltham, MA, USA). The supernatant was discarded and 2 mL of M9 minimal media (*Elbing & Brent, 2002*) was added and the cells were resuspended by gentle vortexing.

### Peptide and antibiotic treatment

Peptides were added as a weight percent of the dry weight of bacterial cells present. Bacterial cell dry weight was calculated in accordance with a previously established standard curve (*Pius, Morrow & Booth, 2012*) relating the OD600 to the dry weight of cells present, and used to calculate the amount of peptide required for a particular treatment. An appropriate amount of a 1 mg/ml stock of AMP, or antibiotic stock solution, either kanamycin or streptomycin (Sigma-Aldrich CO, St. Louis, MO, USA) was added after resuspension of JM109 cells in M9, followed by gentle inversion (at room temperature) to mix the JM109 cells and peptide.

### Differential Scanning Calorimetry (DSC)

Immediately after peptide or antibiotic treatment, the samples and reference solutions were degassed at 4 °C for 5–10 min. M9 minimal media was used for the reference cell. After degassing, the sample and reference were loaded into the cells of a Nano DSC (TA Instruments, New Castle, DE, USA), taking care not to introduce any air bubbles. The DSC and samples were equilibrated during a brief DSC scan from 25 °C to 5 °C. Scans were performed immediately afterward, with a temperature increase from 5 °C to 125 °C at

a rate of 0.5 °C/min, under a pressure of 3 atm. Samples were cooled back to 5 °C and a rescan performed. After completion the DSC cell was washed with 1L of distilled water, 1 L of 2% Contrad, and then 1 L of distilled water.

Raw DSC data was plotted using the program DataGraph (Visual Data Tools Inc.). The water-water baselines for our DSC instrument are linear and the sample rescans are also linear with the exception of a few small peaks. Thus, we opted to subtract linear baselines only from the data, rather than employing a more complicated, and potentially artifact-prone baseline subtraction scheme. The baselines we used were linear fits to the high and low temperature extremes of the plotted data and were subtracted from the raw scans. That is to say, for each temperature range of interest and displayed in the figures, a separate linear fit to the endpoints was performed and subtracted from the scan.

## Minimal inhibitory concentration assays

Minimal Inhibitory Concentration (MIC) assays were carried out using JM109 *Escherichia coli*. The working concentration was $1.6 \times 10^5$ cfu/mL in Mueller Hinton (MH) broth. MSI-78 was dissolved in 20 mM sodium phosphate buffer, pH 7, to give a concentration of 1 mg/mL. 100 µL of MSI-78 solution was added to wells B2 and C2 of an untreated, sterile 96-well polypropylene plate. Peptide concentrations ranged from 250 µg/mL to 0.49 µg/mL. Sodium phosphate buffer controls ($v = 100$ µL) were added to wells D2 and E2. 100 µL of Solution A (0.04% acetic acid +0.02% bovine serum albumin) was added to rows B2-B11, C2-C11, D2-D11, and E2-E11. Serial dilutions were carried out by addition of 100 µL of previous well to subsequent wells in the lettered row (i.e., 100 µL of B2 added into B3). 100 µL of column 11 was discarded to keep same working volume. 100 µL of diluted JM109 (1/200) was added to all wells. Controls included bacteria and MH broth alone. The plates were incubated at 37 °C in a shaking incubator set at 140 RPM for 16 h. The MIC was defined as the lowest concentration of peptide that resulted in no bacterial culture growth as judged by visual inspection of the opacity.

## Hemolytic assay

Hemolytic assays were performed in triplicate to test MSI-78 and scrambled MSI-78 activity against eukaryotic (human) cells. Erythrocytes were obtained from blood collected in a 60 mL conical tube with an EDTA anti-coagulant. This was then transferred to 15 mL conical tubes, and the level of blood was marked with a permanent marker. The tubes were centrifuged at 3,000 rpm at 4 °C for 15 min. The plasma was discarded and erythrocytes were resuspended in 150 mM Tris–HCl buffer to the marked line levels, giving the same concentration of erythrocytes as was present in the previous step. Two more washing steps of centrifugation and cell resuspension were completed. After the final washing, the erythrocytes were resuspended in the appropriate volume of 100 mM sodium phosphate buffer, pH 7. A 1 mL sample of this resulting solution was added to 9 mL of 100 mM sodium phosphate buffer for a dilution of 1:10. This yielded an erythrocyte concentration of approximately $5 \times 10^8$ cells/mL.

Two stock peptide solutions were made: 0.5 mg/mL MSI-78 and scrambled MSI-78. Solutions were made in 100 mM phosphate buffer and appropriate volumes were added to

give the desired concentration in a 1.30 mL sample volume. Then, 200 μL of the diluted erythrocytes was added. The final concentrations of the peptides after these dilutions were 24.35 micrograms/mL and 48.7 micrograms/mL for both peptides that were assayed. Two controls were utilized, one exhibiting 100% hemolysis, and one that exhibits 0%. The positive (100%) control was made by adding 200 μL of diluted erythrocytes to 1.3 mL of 1 % Triton X-100 detergent solution in 100 mM sodium phosphate buffer. The negative (0%) control was made by adding 200 μL of diluted erythrocytes to 1.3 mL of 100 mM sodium phosphate buffer. Solutions were incubated in a water bath for an hour at 37 °C. Samples were then centrifuged at 3,000 rpm, 4 °C for 15 min to pellet any erythrocytes that were not lysed. The supernatants were collected by pipette into cuvettes and measured at 541 nm on a Genesys 10S UV–VIS Thermo Scientific Spectrometer (Thermo Fisher Scientific Inc., Waltham, MA, USA).

Percent hemolysis was calculated by using the equation:

$$\%\text{Hemolysis} = \frac{\text{Absorbance of Sample-Absorbance of Negative Control}}{\text{Absorbance of Positive Control}} \times 100\%. \quad (1)$$

### In vitro transcription/translation assay

*E. coli* JM101 cells were made competent by the rubidium chloride method. These cells were transformed with 4 ng of Luciferase T7 Control DNA plasmid (Promega, Madison, WI, USA) and spread on Luria-Bertani (LB) agar plates containing 100 μg/mL ampicillin. After incubation at 37 °C for 18–24 h, 10 mL of LB broth containing 100 μg/mL ampicillin was inoculated with a single colony from the overnight plates. Plasmids were isolated and purified with the PureYield$^{\text{TM}}$ Plasmid MiniPrep System (Promega, Madison, WI). Plasmid concentration was measured by using a Thermo Nanodrop Spectrophotometer (Thermo Fisher Scientific, Wilmington, DE, USA) set to measure for nucleic acids (dsDNA).

*E. coli* T7 S30 Extract System for Circular DNA (Promega, Madison, WI, USA) was used for production of luciferase from Luciferase T7 Control DNA plasmids. Fresh reaction mixture was prepared by adding T7 extract solution, S30 premix, and a complete amino acid mix (made by adding equal parts of two of the amino acid solutions minus one amino acid) in a 3:4:1 ratio (v/v/v). Reactions were set up in test tubes containing 8 μL of reaction mix, and 1 μL of experimental solution (500 μg/mL MSI-78, 5 mg/mL MSI-78, or 500 μg/mL kanamycin) or 1 μL of sterile deionized water for a positive control. 1 μL of isolated plasmid was then added to reactions and immediately incubated for 5 min at 30 °C in a water bath. After the 5 min incubation, reactions were stopped with the addition of 90 μL of a bovine serum albumin (BSA) solution, containing 1 mg/mL BSA and 2mM dithiothreitol (DTT) in 25 mM Tris–HCl buffer. This 10x dilution was sufficient to stop the reaction by diluting the concentration of reaction components necessary for the production of luciferase. Reaction solutions were placed on ice. Luciferase activity was measured using a Synergy Mx Fluorescence Plate Reader (BioTek, Winooski, VT, USA) set to incubate samples for 10 s with agitation, followed by kinetic measurements with minimum interval with a 2 s measurement delay. Twenty μL of each sample was mixed with 100 μL of Luciferase Assay Reagent (Promega, Madison, WI, USA) in a 96 well transparent microplate. Each sample (sterile deionized water, 50 μg/mL MSI-78, 500 μg/mL MSI-78,

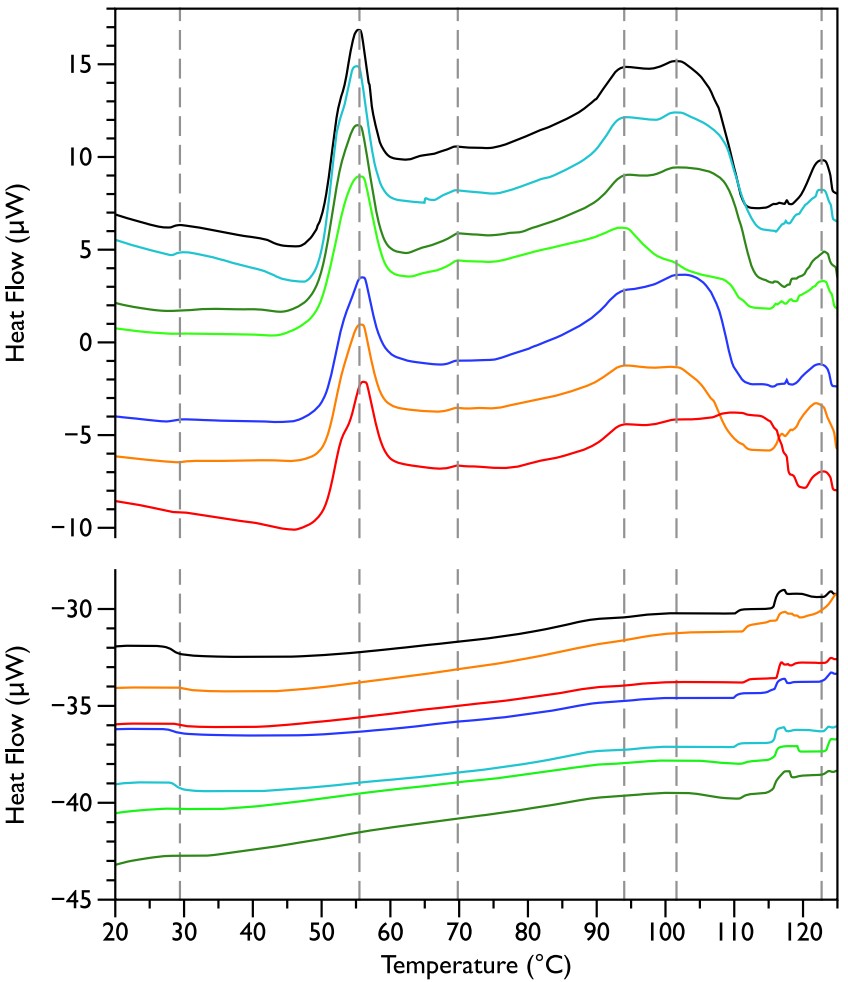

**Figure 1** **Differential Scanning Calorimetry scans of untreated _E. coli_ JM109 cells.** Thermograms for seven different samples, are displayed with an arbitrary vertical offset. The initial scans are shown in the upper panel and the rescans are shown below in the same colour as the first scan. A linear baseline has been subtracted from each of the initial scans but the rescans are shown unmodified. The scan shown in black is the most similar scan to the average of all the control scans and was selected to display in the rest of the figures for comparison. Grey dotted vertical lines indicate peaks that were consistently observed in all initial scans.

and 50 μg/mL kanamycin) was measured in duplicate. The results were normalized by setting the activity in water to 1.0.

## RESULTS

Differential scanning calorimetry was used to assess the thermal transitions in whole _E. coli_ JM109 cells upon addition of increasing concentrations of MSI-78 and small molecule antibiotics. When performing DSC with complex samples such as whole cells, it is important to assess the reproducibility of the scans and thus seven scans of separately prepared untreated bacteria were performed. DSC scans of the bacterial cells alone (Fig. 1, upper panel) exhibit six consistently reproducible endothermic peaks, at temperatures of

29 °C, 55 °C, 70 °C, 94 °C, 102 °C, and 123 °C (indicated by the dashed lines in Fig. 1). These transitions correspond well with those found by other groups (Mackey et al., 1991; Lee & Kaletunc, 2002b). Mackey et al. (1991) identified these transitions with the melting of membrane lipids (20–40 °C), ribosomal components and tRNA (47–85 °C), and DNA and cell wall components at higher temperatures (95–125 °C).

After each bacterial sample was scanned a second time (Fig. 1, lower panel), the only features observed were small peaks at 29 °C and at 117 °C, as well as even smaller peaks at 110 and 125 °C. This is consistent with the Mackey et al. (1991) identification of the 29 °C peak as lipid, as well as their finding that the region from 110 and 125 °C included contributions from DNA and carbohydrate transitions. Lipid transitions, as well as some DNA and carbohydrate transitions, are expected to be reversible, but given the crowded environment present in the whole cells, the proteins are expected to denature irreversibly (Lepock, 2005).

Figure 2 shows thermal scans of JM109 cells treated with increasing amounts of MSI-78, ranging from 2.5% to 60%, where the % expressed is the dry weight of peptide compared to the dry weight of the bacteria (as in Pius, Morrow & Booth, 2012). As expected for a peptide believed to have a lytic mechanism, the addition of MSI-78 does cause the lipid transition at ∼29 °C to disappear, even with the smallest amount of MSI-78, consistent with MSI-78 disruption of the lipid bilayer. However, the major change that occurs in the DSC scans with MSI-78 is a decrease in the amplitude of the 55 °C peak with increasing peptide, and the concomitant appearance of a new peak at 40 °C that increases in amplitude with increasing concentrations of MSI-78. One explanation consistent with this observation is an interaction between MSI-78 and the ribosomes that induces the formation of a less stable ribosome structure that denatures at a lower temperature.

The normal protocol employed for MSI-78 treatment of the cells in this study was to leave the MSI-78 in the samples during the scan—i.e., in general there was no wash step to remove any excess MSI-78 that might remain free in solution rather than bound to the cells. Therefore, two scans were performed in which the cells were treated with MSI-78, but then the cells centrifuged and supernatant discarded to remove any excess MSI-78. The result (Fig. S1) was identical to the scans shown in Fig. 2. Hence, it appears that the MSI-78 is able to interact with the cells before the start of the scan.

In order to probe the specificity of the MSI-78 effects on the DSC scans, the experiments were repeated with a scrambled version of MSI-78—i.e., a peptide with identical overall amino acid composition, but the residue order scrambled so that it cannot form an amphipathic helical structure. Scrambled MSI-78 is 8 times less active against JM109 in MIC assays (Table 1) and also about 8 times less active against human red blood cells (Table 2). In a qualitatively similar manner to MSI-78, the addition of scrambled peptide causes a decrease in the amplitude of the 55 °C peak in the DSC scans of *E. coli* (Fig. 3). However it takes more scrambled peptide to bring about similar effects; 5% scrambled peptide causes slightly less of a decrease in peak size than 2.5% MSI-78. And in further contrast to MSI-78, the scrambled peptide does not induce a concomitant appearance in the peak at 40 °C. At 60% scrambled peptide, an exotherm centered at ∼37 °C is apparent. This exotherm is potentially also present in the MSI-78 scans, but obscured by

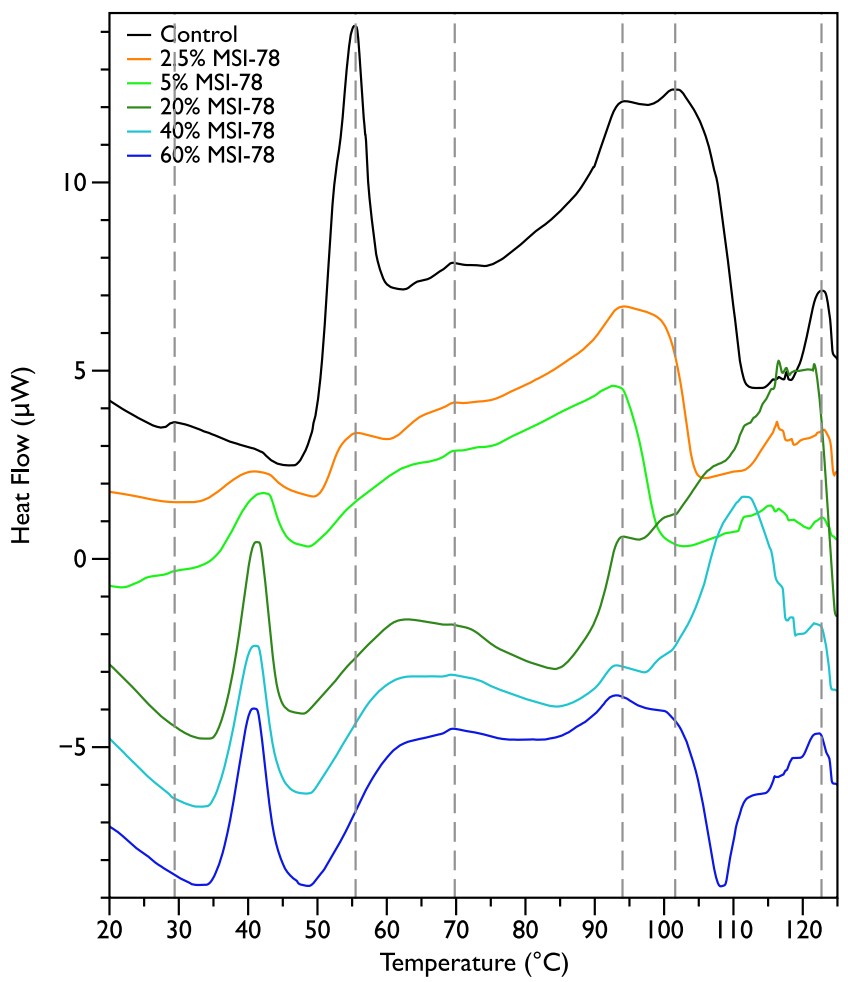

**Figure 2** Differential Scanning Calorimetry scans of *E. coli* JM109 cells treated with the AMP MSI-78. JM109 cells were treated with 0 to 60% weight MSI-78 per dry weight of cells.

**Table 1** Minimal inhibitory concentration (MIC) data for JM109 *E. coli* with MSI-78 and scrambled MSI-78. The experiments were performed in triplicate—identical values for each peptide were obtained each time.

| MSI-78 | Mean MIC against JM109 (ug/ml) |
| --- | --- |
| MSI-78 | 2.0 |
| MSI-78s | 16.5 |

the appearance of the new peak at 40 °C. The exothermic peak at ∼37 °C likely derives from metabolism of the peptide by the bacteria (*Lepock, 2005*).

Since MSI-78 appears to affect the 55 °C DSC peak associated with ribosomes, further experiments were carried out with two antibiotics known to affect the 30S ribosomal unit, namely streptomycin and kanamycin. Streptomycin (Fig. 4A) did not cause any major changes in the DSC peaks compared to the control scans, and in particular caused

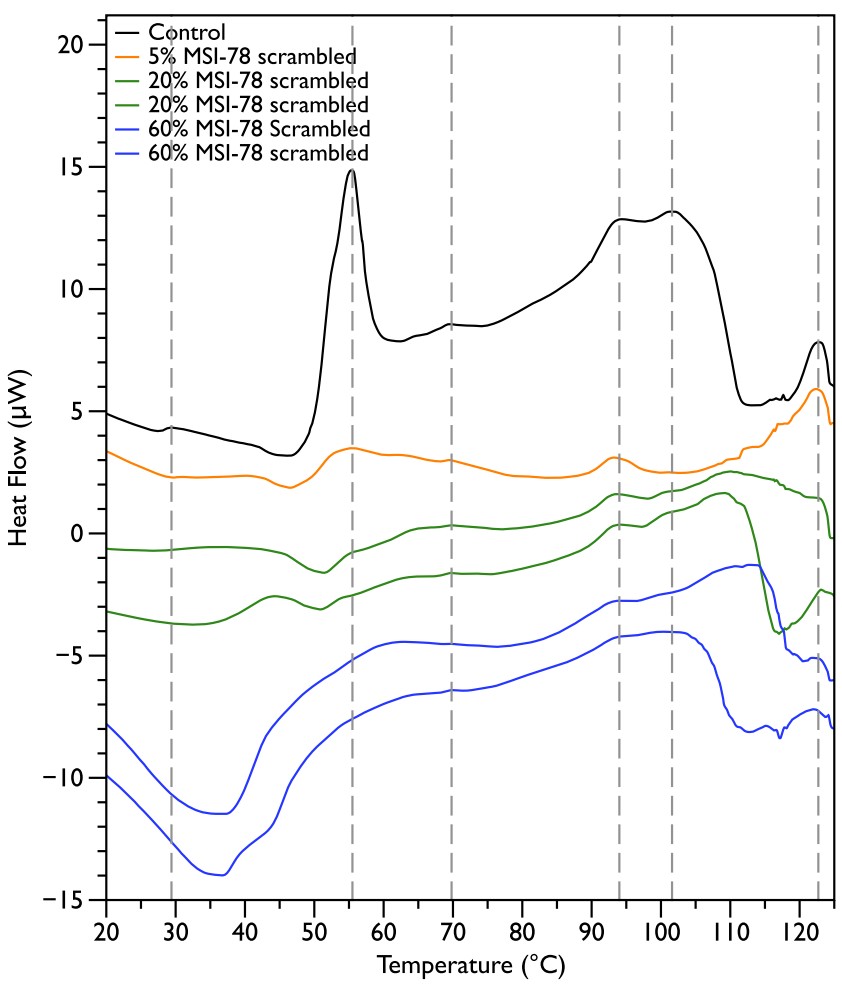

**Figure 3** Differential Scanning Calorimetry scans of *E. coli* JM109 cells with addition of a scrambled variant of MSI-78. JM109 cells were treated with a scrambled variant of MSI-78, i.e., a peptide with the same amino acid content as MSI-78 but the primary sequence order scrambled so it cannot form an amphipathic alpha-helix.

**Table 2** Hemolytic Activity of MSI-78 and scrambled MSI-78. The values indicated are the average of absorbance measurements from 3 samples measured at 541 nm.

| Sample | Average absorbance | Standard deviation | % Hemolysis |
|---|---|---|---|
| Negative control | 0.033 | 0.0039 | N/A |
| Positive control | 0.870 | 0.064 | N/A |
| 24.35 μg/mL MSI-78 | 0.093 | 0.0059 | 6.90% |
| 48.7 μg/mL MSI-78 | 0.116 | 0.010 | 9.54% |
| 24.35 μg/mL scrambled MSI-78 | 0.0517 | 0.0074 | 2.15% |
| 48.7 μg/mL scrambled MSI-78 | 0.0493 | 0.0041 | 1.87% |
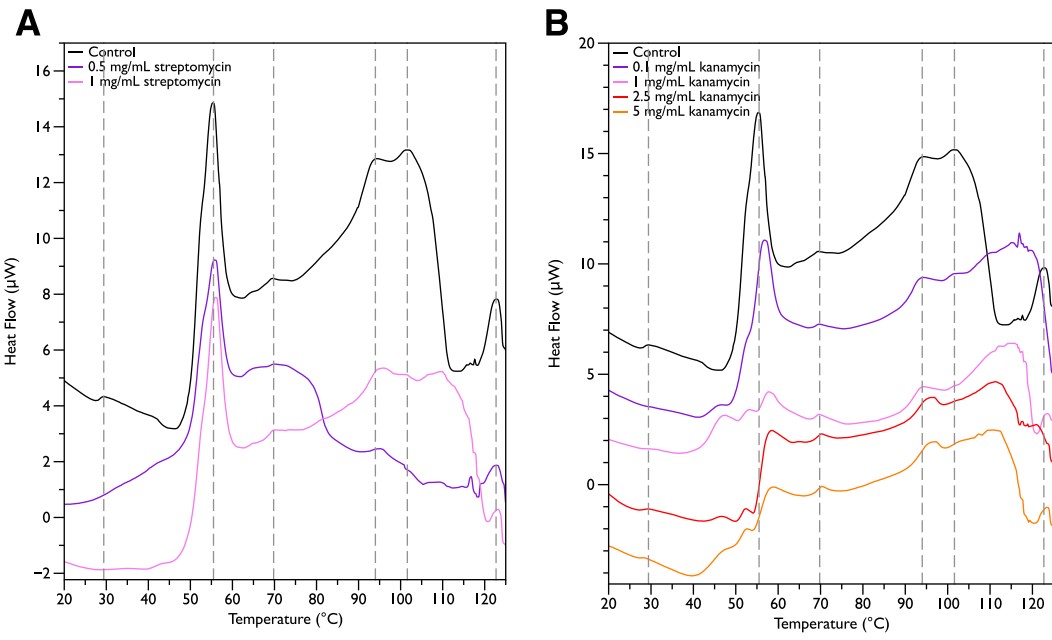

**Figure 4** **Differential Scanning Calorimetry Scans of *E. coli* JM109 cells treated with various concentrations of aminoglycoside antibiotics streptomycin and kanamycin.** JM109 cells were treated with the antibiotic (A) streptomycin, and (B) kanamycin. Both are inhibitors of the bacterial 30S small ribosomal subunit.

no change in the 55 °C peak. With kanamycin (Fig. 4B), there was a large decrease in intensity of the 55 °C peak and a concomitant appearance of small peaks at ∼47, 52, and 57 °C. The new small peaks may correspond to structures with transition temperatures that have been modified by the kanamycin. Or alternatively, at least for the 52 and 57 °C peaks, may correspond to transitions present in the untreated cells but "revealed" by the kanamycin-induced reduction in intensity for the strong 55 °C peak. In any case, it is apparent that not all ribosome-affecting agents have the same effect on the DSC scan of the bacteria. This is perhaps not surprising, given that kanamycin and streptomycin are known to have different effects on the ribosome—e.g., streptomycin decreases the mobility of selected regions of the 30 s unit (*Gromadski & Rodnina, 2004*), whereas kanamycin stabilizes peptidyl-tRNA binding (*Semenkov et al., 1982*).

In order to look for more subtle MSI-78-induced changes in the DSC thermograms, three temperature ranges, from 35 to 60, 68 to 75 and from 90 to 97 °C were analyzed individually. For each of these temperature ranges, a linear baseline was fit to the scan over the lower and upper regions of the temperature range and this baseline subtracted from the data. The untreated bacteria scans were plotted first to give an idea of the reproducibility of the thermograms (Fig. 5). Upon addition of MSI-78 (Fig. 6), changes were seen in the scans in these regions, but, with the exception of the previously discussed 55 °C and 41 °C peaks, the changes were of the same order as the variations between control scans. Additionally, the MSI-78 induced alterations to the DSC scans didn't show a consistent pattern of increase or decrease with increasing peptide concentration. Thus, if there are MSI-78 induced changes in other components of the cell, we cannot detect them using this

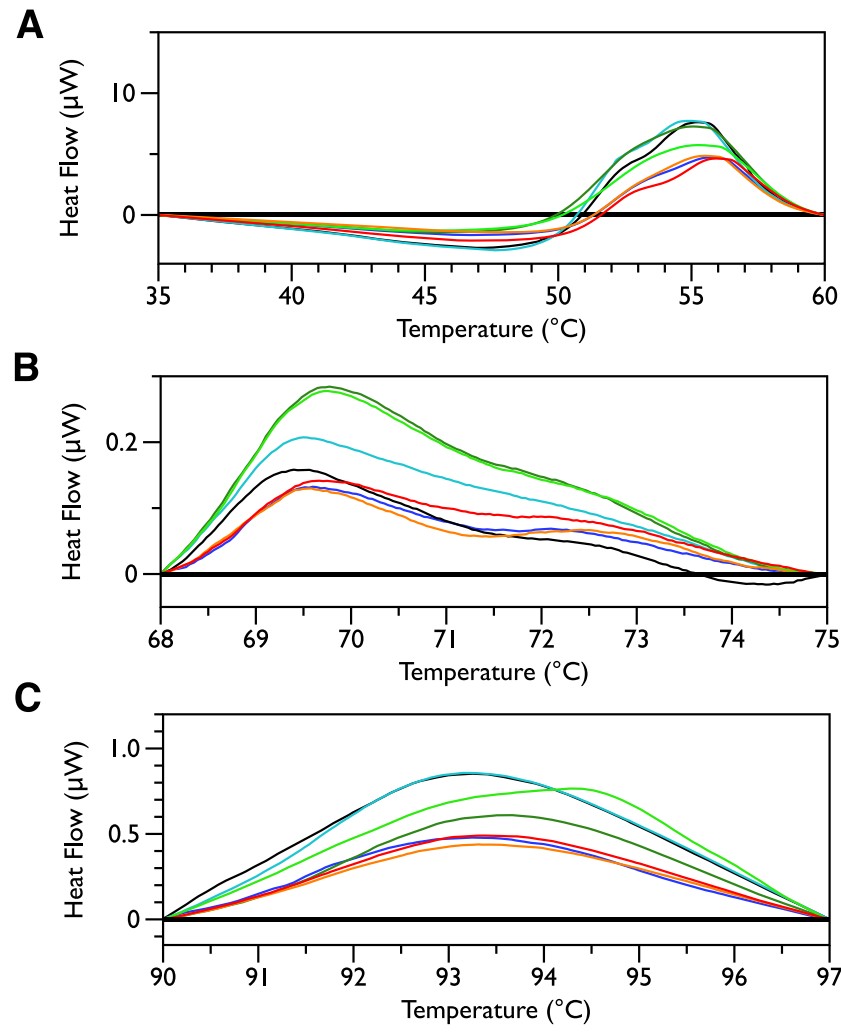

**Figure 5 DSC scans of untreated *E. coli* JM109 cells with temperature ranges analyzed separately.** A linear baseline was fit to each scan over the lower and upper regions of each temperature range and this baseline subtracted from the data (A) 35 to 60 °C, (B) 68 to 75 °C, and (C) 90 to 97 °C.

method. Interestingly, the scrambled peptide reduces the intensity of the peaks in the 68 to 75 and 90 to 97 °C regions in a manner that increases with increasing peptide concentration (Fig. 7). That MSI-78 has greater effects than the scrambled peptide on the 55 °C peak, but smaller effects than the scrambled peptide on the rest of the thermogram, suggests that MSI-78's action is more specific than the scrambled peptide. Neither streptomycin nor kanamycin showed consistent changes to the 68 to 75 and 90 to 97 °C regions (Figs. 4A and 4B).

To provide further evidence of MSI-78's effects on ribosomes, an *in vitro* transcription/translation assay (*Pedrolli & Mack, 2014*) was performed. MSI-78 at 50 ug/mL inhibited the synthesis of luciferase by extracted ribosomes by ∼50% and 500 ug/mL MSI-78 knocked the activity down to the same level as 50 ug/mL kanamycin (Fig. 8).

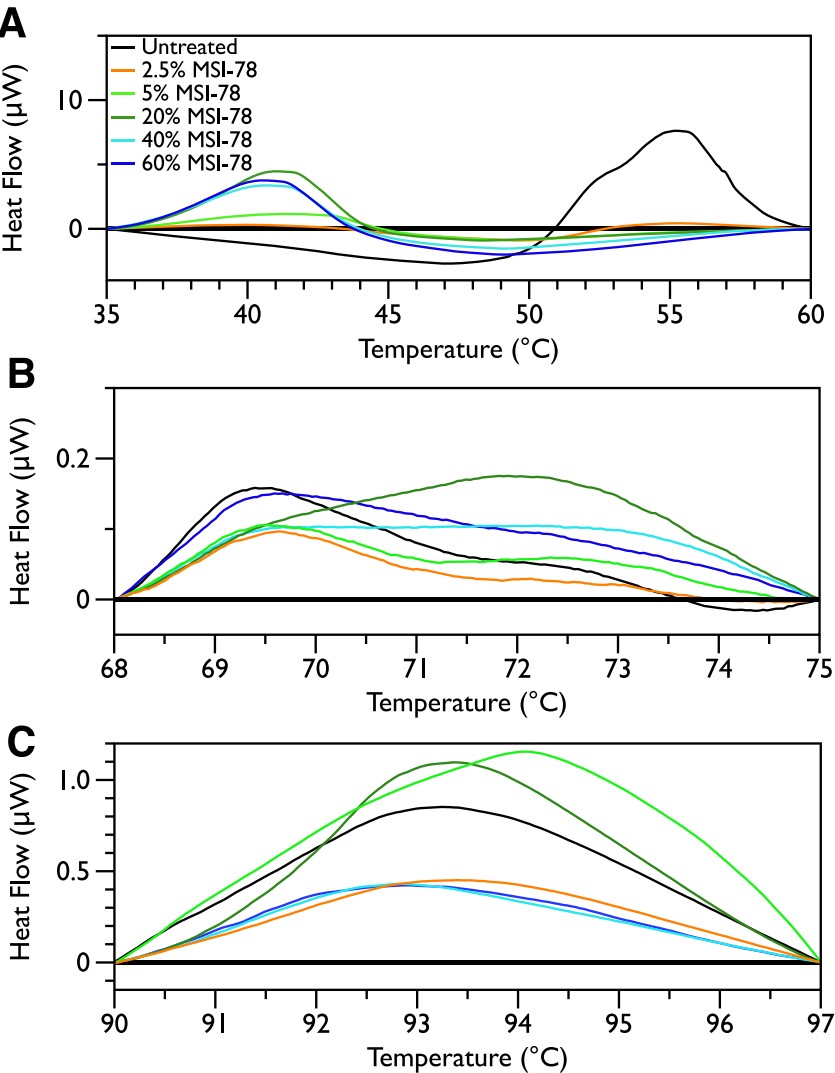

**Figure 6  DSC Scans of *E. coli* JM109 Cells treated with MSI-78 with temperature ranges analyzed separately.**  A linear baseline was fit to each scan over the lower and upper regions of each temperature range and this baseline subtracted from the data (A) 35 to 60 °C, (B) 68 to 75 °C, and (C) 90 to 97 °C. The colours match those from Fig. 2.

## DISCUSSION

The goal of this study was to probe potentially functional interactions of the AMP MSI-78 with non-lipidic extracellular and intracellular components of intact *E. coli*, as a step towards a more global understanding of how AMPs target and inhibit pathogen growth. Using differential scanning calorimetry (DSC) of whole *E. coli* treated with MSI-78 we were able to show that, thermodynamically speaking, MSI-78's most striking effect on the bacterial cells is on the ribosomes (Fig. 2). Increasing concentrations of MSI-78 caused a decrease in the area under the ribosome peak with the concomitant appearance and growth of a peak at a lower temperature, consistent with an MSI-78-induced destabilization of the ribosome. This DSC identification of ribosomes as an MSI-78 target was supported by

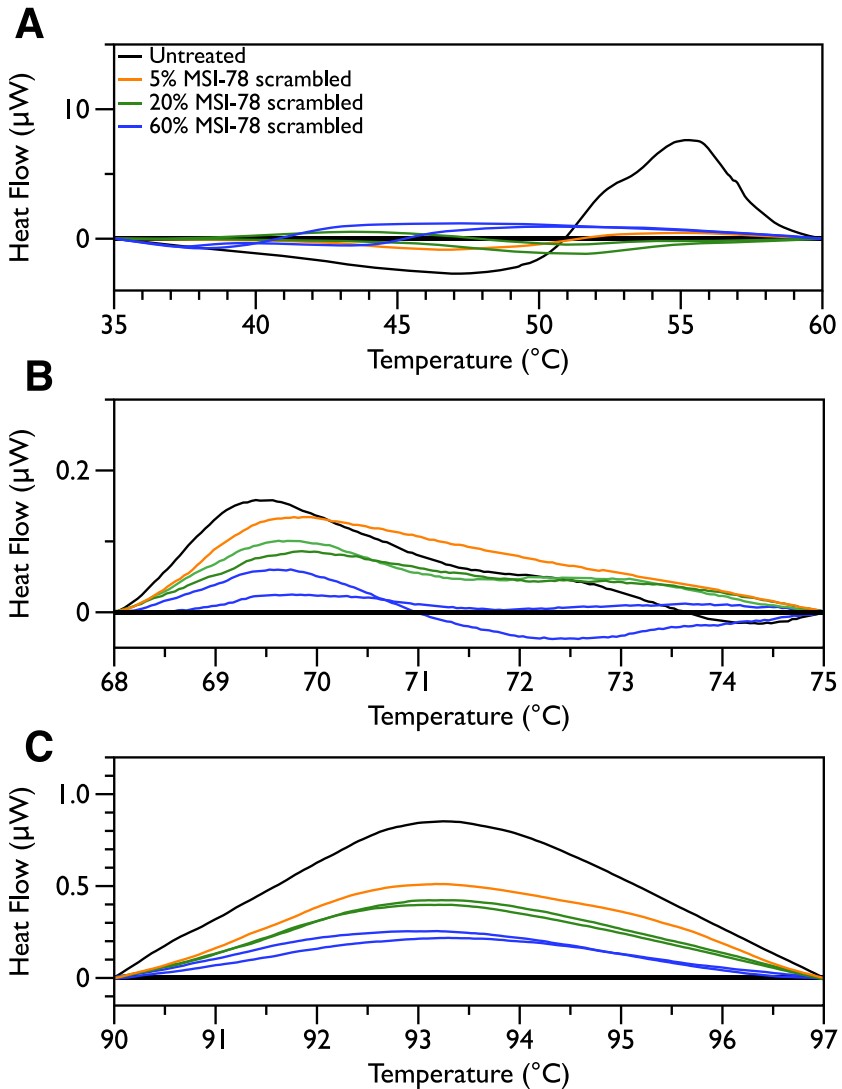

**Figure 7  DSC Scans of *E. coli* JM109 Cells treated with scrambled MSI-78 with temperature ranges analyzed separately.** A linear baseline was fit to each scan over the lower and upper regions of each temperature range and this baseline subtracted from the data (A) 35 to 60 °C, (B) 68 to 75 °C, and (C) 90 to 97 °C. The colours match those from Fig. 3.

the results of an *in vitro* transcription/translation assay (Fig. 8). The data available from this study are insufficient to identify a particular mechanism by which MSI-78 disrupts ribosome function, but one possibility is that MSI-78 binds to a target in the ribosome complex and blocks the interaction with translation substrates or induces a ribosome conformation that is less able to carry out translation. While interfering with ribosome function could certainly lead to growth inhibition, it is not possible to be certain from the DSC data if the disruption caused by MSI-78 is sufficient to cause growth inhibition. In terms of AMP:lipid ratios, the amounts of AMP employed in the DSC scans could be expected to be in the sublethal range. However, given the disparate experimental conditions between MIC assays and the DSC work, such as cell density (and detailed in the

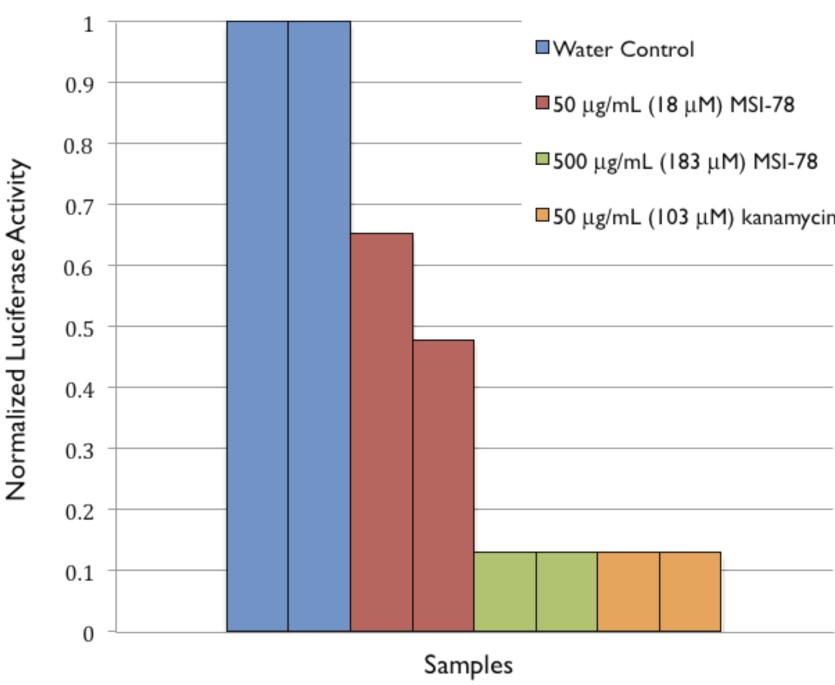

**Figure 8** **Normalized Luciferase Activity of** *in vitro* **Transcription/Translation Assay of Vector-bound Luciferase to Test for Inhibition of Protein Synthesis.**

Introduction), AMP:lipid ratio may not provide a meaningful comparison. There is indeed precedence for AMP to function via interference with ribosomes, for example the AMP PR-39 (*Boman, Agerberth & Boman, 1993*) has been shown to inhibit protein synthesis. Furthermore, magainin 2, the AMP on which the MSI-78 sequence is based, has been shown to locate to the cell cytoplasm (*Haukland et al., 2001*).

In contrast to MSI-78, a scrambled version of MSI-78 showed a weaker effect (Fig. 3) on ribosomes, and unlike MSI-78, also affected other cellular components (Fig. 7). Together, the results with MSI-78 and its scrambled version suggest that MSI-78's effects are at least somewhat dependent on its particular structural features, rather than a sole function of its overall charge and hydrophobicity. This finding is consistent with earlier studies that showed that disrupting the secondary structure of MSI-78's related AMP, magainin-2, and analogs also disrupted their function (*Chen et al., 1988*; *Matsuzaki et al., 1997*). One of the hopes of our study was to examine if it might be possible to use DSC to see interactions between MSI-78 and the carbohydrate components of the cell envelope. However, our results contained too much sample-to-sample variation in this region of the DSC scans to say definitively if MSI-78 had an effect on these components or not. It would be interesting in future to perform DSC scans of cells treated with a variety of AMPs with previously identified targets including other well-studied AMPs that are thought to be lytic, as well as AMPs that target DNA or cell envelope components.

Our earlier NMR work with MSI-78 and whole cells suggested that, in addition to its well characterized lipid interactions, MSI-78 interacts with non-lipidic cell envelope components and intracellular targets (*Pius, Morrow & Booth, 2012*). The current study with

DSC verifies that MSI-78 does indeed have an intracellular target and that this target is the ribosomes. MSI-78's ability to permeabilize model and *E. coli* membranes has previously been well documented ((*Hallock, Lee & Ramamoorthy, 2003*; *Ramamoorthy et al., 2006*); *Gottler & Ramamoorthy, 2009*; *Lee et al., 2013*). Putting all the results together, it appears that MSI-78 acts via multiple mechanisms—i.e., by inhibiting ribosomal activity as well as disrupting the cell membrane. Multiple researchers have suggested that at least some AMPs kill target pathogens via a multi-hit mechanism (e.g., *Friedrich et al., 2000*; *Hancock & Sahl, 2006*; *Jenssen, Hamill & Hancock, 2006*; *Marcos & Gandia, 2009*; *Wimley, 2010*; *Nguyen, Haney & Vogel, 2011*) and the DSC work is certainly consistent with a multiple-hit mechanism for MSI-78. Our study only examined one AMP, and so extending the results to other AMPs would be speculative. However, what it not speculative is the observation that once an AMP has been found to lyse model or pathogenic membranes, it is rare for that AMP to be subjected to further analysis to identify potential non-lipidic mechanisms of action. One potential consequence of this narrow focus with respect to an AMP's mechanism is the difficulty encountered in rational design of AMPs for clinical use via optimization of AMP-lipid interactions. The poor correlation between model membrane lytic activity and effectiveness in MIC assays observed for some AMPs (e.g., *Friedrich et al., 2000*; *He, Krauson & Wimley, 2014*) may well be due to the presence of non-lipid targets of an AMP that are not reflected in the model membranes used for AMP optimization. Successful optimization of an AMP for clinical use may well require optimization of AMP interactions with both lipid and non-lipid targets of that AMP.

## ACKNOWLEDGEMENTS

The authors would like to express their gratitude to Matthias Mack, for his advice on the transcription assay, Rob Brown for help with the hemolytic assays, and Kapil Tahlan for fruitful discussions.

### Funding

This research was supported by an NSERC Discovery Grant to V. Booth (312676). The funders had no role in study design, data collection and analysis, decision to publish, or preparation of the manuscript.

### Grant Disclosures

The following grant information was disclosed by the authors:
NSERC Discovery: 312676.

### Competing Interests

The authors declare there are no competing interests.

## Author Contributions

- Alexander M. Brannan conceived and designed the experiments, performed the experiments, analyzed the data, wrote the paper, prepared figures and/or tables, reviewed drafts of the paper.
- William A. Whelan and Emma Cole performed the experiments, analyzed the data, reviewed drafts of the paper.
- Valerie Booth conceived and designed the experiments, analyzed the data, wrote the paper, prepared figures and/or tables, reviewed drafts of the paper.

## Data Availability

    Raw data can be found in the Supplemental Information.

## Supplemental Information

Supplemental information for this article can be found online at http://dx.doi.org/10.7717/peerj.1516#supplemental-information.

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
