# Peer review of "Differential scanning calorimetry of whole Escherichia coli treated with the antimicrobial peptide MSI-78 indicate a multi-hit mechanism with ribosomes as a novel target"

_PeerJ, doi:10.7717/peerj.1516_

## Round 0.1 · original submission · Minor Revisions

In addition to the detailed comments from the reviewers, the authors should make the following changes:

1) Gottler et al reference is duplicated as 2009a and 2009b. Instead, Solid State Nucl. Magn. Reson. 35 (2009) 201 can be included.
2) Biochemistry, 2015, 54 (10), pp 1897, and Biochemistry 45 (2006) 5793 and reported more mechanistic insights into MSI-78 and related peptides. These should be included.

Reviewer 1 ·

Basic reporting

Brannan et al. describe the interaction study of novel antimicrobial peptide MSI 78 with whole cell using DSC to understand the mechanism of antimicrobial peptide targeting to Ribosome.

Experimental design

Nicely described

Validity of the findings

Nice findings and good presentation.

Additional comments

This is a nice study; presentation of the data, the conclusion drawn from it and the discussion of the implications are of high level. I suggest the authors clarify few following points:

1. Line 8 “Known non-lytic mechanisms for AMPs include targeting nucleic acids (Park et al., 1998), chaperones (Kragol et al., 2001),……….”

Authors can include few references in this regard: Mol. Biosys. 9(7), 1833-6 9(7):1833-6, ChemMedChem. 9(9), 2059-64 etc.


2. Line 307 “Since MSI-78 appears to affect the 55°C DSC peak associated with ribosomes, 308 further experiments were carried out with two antibiotics known to affect the 30s 309 ribosomal unit, namely streptomycin and kanamycin…………’’

Since the mechanism of action of AMPs and antibiotics are not similar, authors should perform/discuss the same experiment with another well-known AMP (s) that targets lipid bilayer or/and Ribosomes in support of their results.

3. Also authors may perform/discuss another DSC experiment with AMP that target to DNA and cell wall components (which may affect the peak at high temp. as mentioned in line 263) to validate experimental results.

Annotated reviews are not available for download in order to protect the identity of reviewers who chose to remain anonymous.

Reviewer 2 ·

Basic reporting

In this study, the authors report use of DSC to probe into intracellular targets of antimicrobial peptides (AMPs). The result suggests that AMP MSI-78 interacts with ribosomes, which may contribute to the antimicrobial effect. The DSC experiments and data analysis were carefully conducted, and the conclusion seems to be supported by the data. However, it is not clear how the impact to ribosomes is related to the antibacterial peptide mechanism and cell viability.

Comments.
1) Ribosome-peptide interaction. The result indicates that the peptide interacts with ribosomes and changes the thermodynamic property of ribosomes. It would be more convincing if this can be elaborated more. What is the possible molecular change in ribosomes by the peptide? How dose this change impact the cellular activity, leading to bacterial growth inhibition or cell death?

2) Cell viability. The peptide concentrations were given by wt. % of peptide relative to total bacteria weight. At these peptide concentrations, what are the cell viabilities of bacteria or how many bacteria are killed by the peptide? Are they higher or lower than MIC? Is this peptide activity bacteriostatic or bactericidal? These information would be critical to know the relationship between the changes in the DSC scans and bactericidal/bacteriostatic mechanisms medicated by peptide-ribosome interaction. While the result indicates that the peptide interacts with ribosomes, the peptide may be able to inhibit bacterial growth without targeting ribosomes.

3) New peak at 40C. Line 281 “This observation is consistent with an interaction between MSI-78 and the ribosomes that induces the formation of a less stable ribosome structure that denatures at a lower temperature.” How do you know the new peak is resulted from a less stable ribosome structure? This seems to be speculation, rather than conclusion.

4) Membrane-active benchmark. Only antibiotics that act in ribosomes are tested. It would provide strong support if membrane-active agents such as surfactants are tested as a control to see how membrane disruption changes DSC scans.

Experimental design

No Comments

Validity of the findings

No Comments

Additional comments

No Comments

---

## Round 0.2 · accepted · Accept

The revised manuscript is ready for publication.